# Monitoring physical behavior in pediatric physical therapy: A mixed methods feasibility study to evaluate a newly developed toolkit and training

Barbara Engels[ID][1,2,3☯], Elles Kotte[4☯], Raoul Engelbert[ID][5,6], Marleen E. Sol[ID][1,2], Remko van der Lugt[ID][7], Harriët Wittink[ID][1], Jan Willem Gorter[ID][8,9], Manon A. T. Bloemen[ID][1,2*]

**1** Research Centre Healthy and Sustainable Living, Research Group Lifestyle and Health, Utrecht University of Applied Sciences, Utrecht, The Netherlands, **2** Research Centre Healthy and Sustainable Living, Research Group Moving, Growing, and Thriving Together, Utrecht, The Netherlands, **3** UMC Utrecht Brain Center and Center of Excellence for Rehabilitation Medicine, Utrecht University, Utrecht, The Netherlands, **4** Fitkids Foundation, Amsterdam, The Netherlands, **5** Centre of Expertise Urban Vitality, Faculty of Health, Amsterdam University of Applied Sciences, Amsterdam, The Netherlands, **6** Department of Rehabilitation Medicine, Amsterdam UMC, University of Amsterdam, Amsterdam, The Netherlands, **7** Research Centre for Learning and Innovation, Research Group Co-Design, Utrecht University of Applied Sciences, Utrecht, The Netherlands, **8** Department of Rehabilitation, Physical Therapy Science and Sports, UMC Utrecht Brain Center, University Medical Center Utrecht, Utrecht, The Netherlands, **9** CanChild, Department of Pediatrics, McMaster University, Hamilton, Ontario, Canada

☯ These authors contributed equally to this work.
* Manon.bloemen@hu.nl

## Abstract

### Introduction

Pediatric physical therapists (PPTs) aim to enhance active physical behavior but lack feasible accelerometry devices to assess and evaluate physical activity (PA). We developed an activity monitoring prototype toolkit (AM-p Toolkit) consisting of a wearable, a docking station, a digital tool for data analysis, and physical tools for communication with children and parents. A training for PPTs was also created. We aim to explore the feasibility of the AM-p Toolkit from the perspectives of PPTs, children, and parents and to assess if training improved PPTs' knowledge, skills, and confidence in using the Toolkit.

### Participants and methods

Using an explanatory sequential mixed methods design, we collected data through questionnaires, individual interviews, and focus groups, guided by Bowen's dimensions of 'acceptability,' 'demand,' and 'practicality.' We included children with the ability to walk, their parents, and their PPTs. The training was evaluated by analyzing PPTs' knowledge, skills, and confidence using the AM-p Toolkit. Quantitative results were analyzed descriptively (mean [SD] and median [interquartile range] when appropriate and qualitative data were analyzed thematically.

**Data availability statement:** Data cannot be shared publicly because of sensitive data in interviews which cannot be anonymized completely. Data contains potentially identifying or sensitive patient information as we interviewed children and parents. The Ethical Committee of Research in Health of the University of Applied Sciences Utrecht approved the research proposal. Data are available from the ResearchDrive at HU University of Applied Sciences, Utrecht (contact via onderzoek-support@hu.nl) at reasonable request for researchers who meet the criteria for access to confidential data.

**Funding:** M.A.T. Bloemen - SIA Raak. MKB12.002 - Foundation Innovation Alliance – Regional Attention and Action for Knowledge Circulation - https://regieorgaan-sia.nl/financiering/raak-mkb/ - The funder did not play any role in the study design, data collection and analysis, decision to publish, or preparation of the manuscript.

## Results

Fifteen PPTs, 17 parents, and 20 children completed the study. PPTs rated overall satisfaction on a 10-point scale with the AM-p Toolkit at 6.3 (SD 1.2), and parents rated it 7.3 (SD 1.6). The following themes emerged for acceptability, demand, and practicality respectively: for acceptability: 1) expected added value, 2) quality and usability, and 3) design; for demand: 1) use and non-use, 2) further development, and 3) willingness for future use; and for practicality: 1) time constraints and 2) integration.

## Conclusion

The AM-p Toolkit shows promise in PPT, with generally positive acceptability among all end-users. PPTs see potential for certain groups of children who can benefit from the AM-p Toolkit. Practicality requires improvements in the web application and refinement of the strap. Training is important and can be strengthened by emphasizing the analysis of assessment results, clinical reasoning, and functional goal-setting.

## Introduction

Less than half of the healthy Dutch people aged four years and older reached the national recommendations for overall physical activity behavior in 2022 [1]. Lack of sufficient physical activity is an urgent worldwide problem as it is a leading risk factor for numerous chronic diseases and harms mental health and quality of life [2–5]. For these reasons, both individual-level interventions and public health initiatives aimed at increasing participation in active physical behavior need careful attention [6].

In 2020, the Dutch Olympic Committee-Dutch Sports Federation (NOC*NSF) joined forces with the association of cooperative Health Foundations in their ambition to enable Dutch youth to become the healthiest in the world in 2040 [7]. The Dutch Association for Pediatric Physical Therapy (NVFK) stated in 2021 that this ambition should also apply to children with chronic conditions and consequent disabilities in the Netherlands, emphasizing the inclusion of this group to become the healthiest in the world [8]. Pediatric physical therapists (PPTs) have been identified as ideal conduits to promote active physical behavior in children with and without disabilities [9–11]. Considering the significant disparities in physical activity (PA) participation rates between children with and without disabilities [12–15], significant steps must be taken to let children with chronic conditions and consequent disabilities fully participate in everyday PA similar to their peers without a disability [16,17]. Concerning PPT treatment, a shift is needed towards more participation-based pediatric physical therapy in the child's context to enhance PA participation [11,18–21]. In addition, insight into actual time spent in PA would undoubtedly add value to adopting more active lifestyles, as parents and children are often unaware of their actual PA levels [22]. They may, therefore, not identify a need to become more physically active.

For many years, accelerometry has been recognized in research settings as an accurate and reliable method to assess PA in children and adolescents with and without disabilities [2,23]. Accelerometry opens doors to creating awareness about PA and highlights discrepancies between perceived PA levels and those measured by a device [24]. This awareness ensures more optimal counseling opportunities in PPT practice [25]. There is, however, a significant lack in the availability of validated and feasible single-sensor methods that can easily recognize active physical behavior in children with and without developmental disabilities in everyday PPT practice [26,27].

Our research group has recently validated a prototype activity monitor (AM-p) in children with and without developmental disabilities who are both ambulatory and children who use a wheelchair in daily life [28,29]. It can distinguish between three different types of activities (stationary behavior, locomotion behavior, cycling) [28,29]. We have explored the needs and requirements of end-users such as PPTs, children, and parents to use the AM-p as an assessment tool [27]. It became clear that concomitant tools are needed, such as a suitable and child-friendly strap to fixate the AM-p, information material, and an easy-to-use application for reading and interpreting results [27]. We used this information to develop concomitant tools aligned with end-user needs to facilitate the successful use of the AM-p in PPT practice. We applied a participatory design approach (co-design) during the development process, which generated insights for intervention prototypes [30,31].

The objectives of our study are 1) To explore the feasibility (acceptability, demand, practicality) of the AM-p Toolkit in PPT practice from the perspectives of PPTs, children, and parents to gather feedback for further refinement, and 2) To evaluate whether the content and design of the training align with the needs of PPTs and assess how the training affects their knowledge, skills, and confidence using the AM-p Toolkit.

## Materials and methods

### Study design and ethics

We used the Bowen framework to assess the AM-p Toolkit's feasibility [32]. We chose the three key domains 'acceptability,' 'demand,' and 'practicality,' because they add the most value to the development of the AM-p Toolkit at this stage [33]. The remaining areas are related to practical implementation and are outside the scope of this research. The training was evaluated using four indicators: 1) knowledge, 2) skills, 3) confidence of PPTs in using the AM-p Toolkit in PPT practice, and 4) whether the design of the training aligned with the needs of the PPTs. We used an explanatory-sequential mixed methods design that included a quantitative and sequential qualitative part [34]. In the study's first phase, we collected and analyzed quantitative data on the AM-p Toolkit and the training. In the second phase, we gathered qualitative data to expand upon the initial findings. This included focus groups with PPTs and semi-structured individual interviews with parents and children, guided by a topic list. All group and individual interviews were led by a trained researcher who received the support of observers.

For the quantitative data collection, we generated online questionnaires using Microsoft Forms, creating exploratory questionnaires that targeted specific aspects of the AM-p Toolkit and the training. We pilot-tested the draft questionnaires with PPTs, parents, and children to ensure clarity and face validity. Based on the feedback, we simplified the language in the questionnaires of children and parents. The questionnaires related to the AM-p Toolkit were based on 5-point rating scales to assess the framework of acceptability, demand, and practicality. The interpretation of values is: 1 – strongly disagree; 2 – disagree; 3 – neutral; 4 – agree; 5 – strongly agree. The responses for children were rated with faces ranging from sadness (1) to smiling (5). On a 10-point rating scale, PPTs and parents could rate their overall satisfaction with the AM-p Toolkit. The interpretation of values is: 1 – extremely dissatisfied; 2 – very dissatisfied; 3 – dissatisfied; 4 – somewhat dissatisfied; 5 – neutral; 6 – slightly satisfied; 7 – moderately satisfied; 8 – satisfied; 9 – very satisfied; 10 – extremely satisfied. The questionnaires related to the training also contained 5-point rating scales related to knowledge, skills, and confidence, as well as a 10-point rating scale about the overall satisfaction of the training. The Ethical Committee for Research and Health at the University of Applied Sciences Utrecht, the Netherlands, provided a positive assessment of this study, advising that it is exempt from the Dutch Medical Research Involving Human Subjects Act (file number 155-000-2021).

## Participants and data collection procedure

We defined three participant groups: 1) PPTs, 2) children (6–18 years old), and 3) their parents. We aimed to include 20 PPTs. We recruited PPTs with flyers using social media sites such as LinkedIn and Facebook, contacts of researchers from the University of Applied Science Utrecht, and Fitkids, a nationwide exercise therapy program in the Netherlands [35]. PPTs were eligible to participate if they had not participated in co-designing the AM-p Toolkit and treated children with goals related to a physically active lifestyle.

A team of six trained researchers (MS, BE, AJ, MW, KR, JvE) collected the data under the research team's supervision. Figs 1 and 2 show the timelines of our data collection for PPTs, children and parents, respectively. After the training (January to February 2022), participating PPTs were instructed to use the AM-p Toolkit in at least three children with PA treatment goals for four months (February to May 2022). The following inclusion criteria for children were established: (1) between 6–17 years of age, (2) ambulatory, (3) capable of reading and understanding the Dutch language, (4) ability to understand and follow instructions and study procedures. Using purposive sampling related to disability, age, and gender, we selected children and parents to participate in individual interviews.

PPTs, children, parents or caregivers received written and verbal information about the study, along with an informed consent form. Written informed consent was obtained from PPTs and parents or caregivers. For children under 12, assent was obtained from the child, and parents or caregivers provided written consent. For children aged 12 to 16, both assent

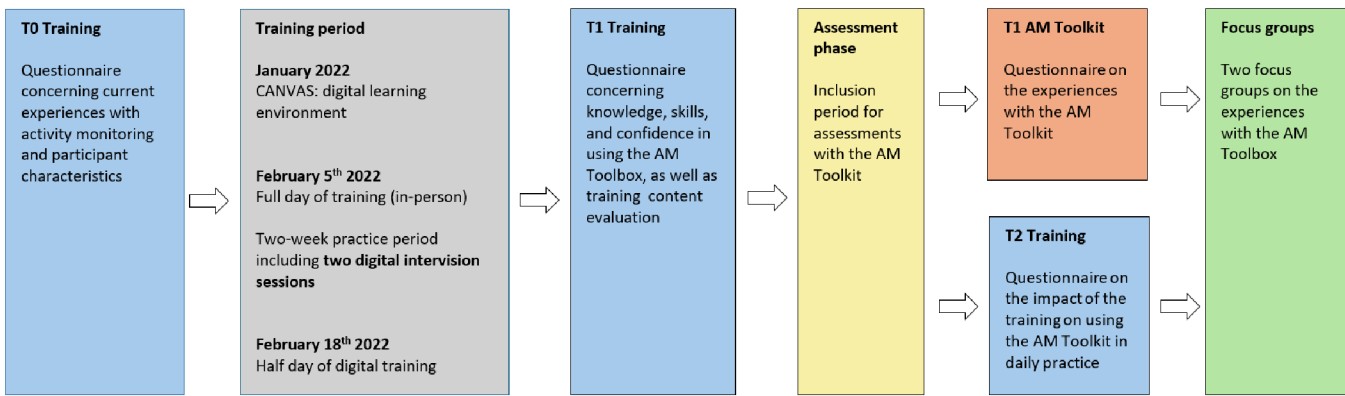

**Fig 1. Timeline data collection pediatric physical therapists (PPTs).** In blue, they completed the T0, T1 and T2 questionnaire focusing on their experiences with the training. In grey, the training period of the blended training is described. In yellow, PPTs included and assessed individual children with the AM-p Toolkit. In green, some PPTs were interviewed in focus groups about their experiences with the AM-p Toolkit and training.

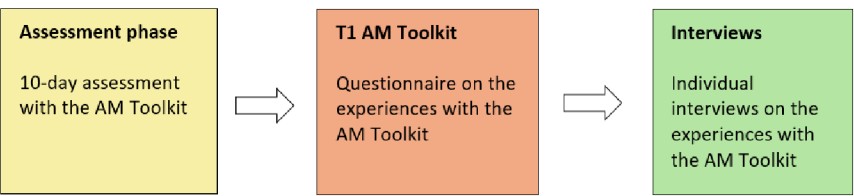

**Fig 2. Timeline data collection children and parents.** In yellow, children (and parents) conducted the 10-day assessment phase with the AM-p Toolkit. After the assessment (orange), they completed the T1 questionnaire focusing on their experiences regarding the AM-p Toolkit. In green, some children and parents were interviewed individually about their experiences with the AM-p Toolkit.

and separate written consent from parents or caregivers were given. From age 16, written consent was obtained directly from the child, provided they had appropriate cognitive development.

## AM-p Toolkit and training

The co-design method applied during the developmental process of the AM-p Toolkit contained (1) three co-creation sessions, (2) two one-week design sprints, (3) living lab testing, and (4) triangulation sessions. Throughout all stages of development, end-users, designers, developers, researchers, and experts from different backgrounds related to PA in childhood and technology were involved. The design process led to the development of an AM-p Toolkit (Fig 3) containing four physical tools. The first tool is a 'docking station' to upload assessment data and recharge the AM-p. The docking station featured a USB port connecting the wearable device, allowing for data upload and charging when plugged in. It also had a reward system of magnets, enabling children to track each wearing day throughout a 10-day assessment period. The AM-p Toolkit also contained 'the wearable,' consisting of an ankle strap and the AM-p. The ankle strap was designed to protect the AM-p device and securely attach it to the ankle. Participants were instructed always to keep the AM-p device in the strap. As a third tool, an information booklet was provided with instructions for parents and children on how to connect and wear 'the wearable.' The final physical tool was an infographic, which the PPT could use to present the assessment results to the child and their parents. A digital tool was also developed: a web application where PPTs could register patients and analyze the assessment data.

The blended training for PPTs included a digital learning environment (CANVAS), a full 8-hour day of training (in-person), a two-week practice period including two digital intervision sessions, and finally, a half-day of online training. All participants received a link to access the digital learning environment containing instruction videos about the practical use of the AM-p Toolkit. These videos served as preparation for the first in-person training. The in-person training aimed to practice hands-on with all the tools: connecting the docking station, creating a user account, wearing 'the wearable,' uploading data, and reading and interpreting results. Following this, PPTs had two weeks to practice actively without patients. In this period, the research team provided two digital intervision sessions, which PPTs could attend if needed. During the half-day online training session, case studies and personal experiences were discussed, and many practical questions were addressed. Between the training days, the research team was reachable for further questions.

## Data analysis

We used IBM SPSS Statistics for Windows, version 27.0 (IBM Corp, Armonk, New York) to analyze the quantitative data, we present the median, interquartile range (IQR), and distribution of ratings for skewed data. We assessed the data's normal distribution using the Shapiro-Wilk test. For normally distributed data, we present the mean and standard deviation (SD). For skewed data, we present the median, interquartile range (IQR), and rating distribution. Additionally, we display the number of participant reactions for each rating on the 5-point rating scale- and 10-point scales.

Focus group and interview data were verbatim transcribed using Amberscript (Amberscript B.V., 2021, San Francisco & Amsterdam), then checked and anonymized by the research team. We included participants for focus groups and interviews until thematic data saturation was reached [36]. Data analysis followed a combined inductive and deductive approach [37]. First, we applied codes to relevant segments using an inductive approach,

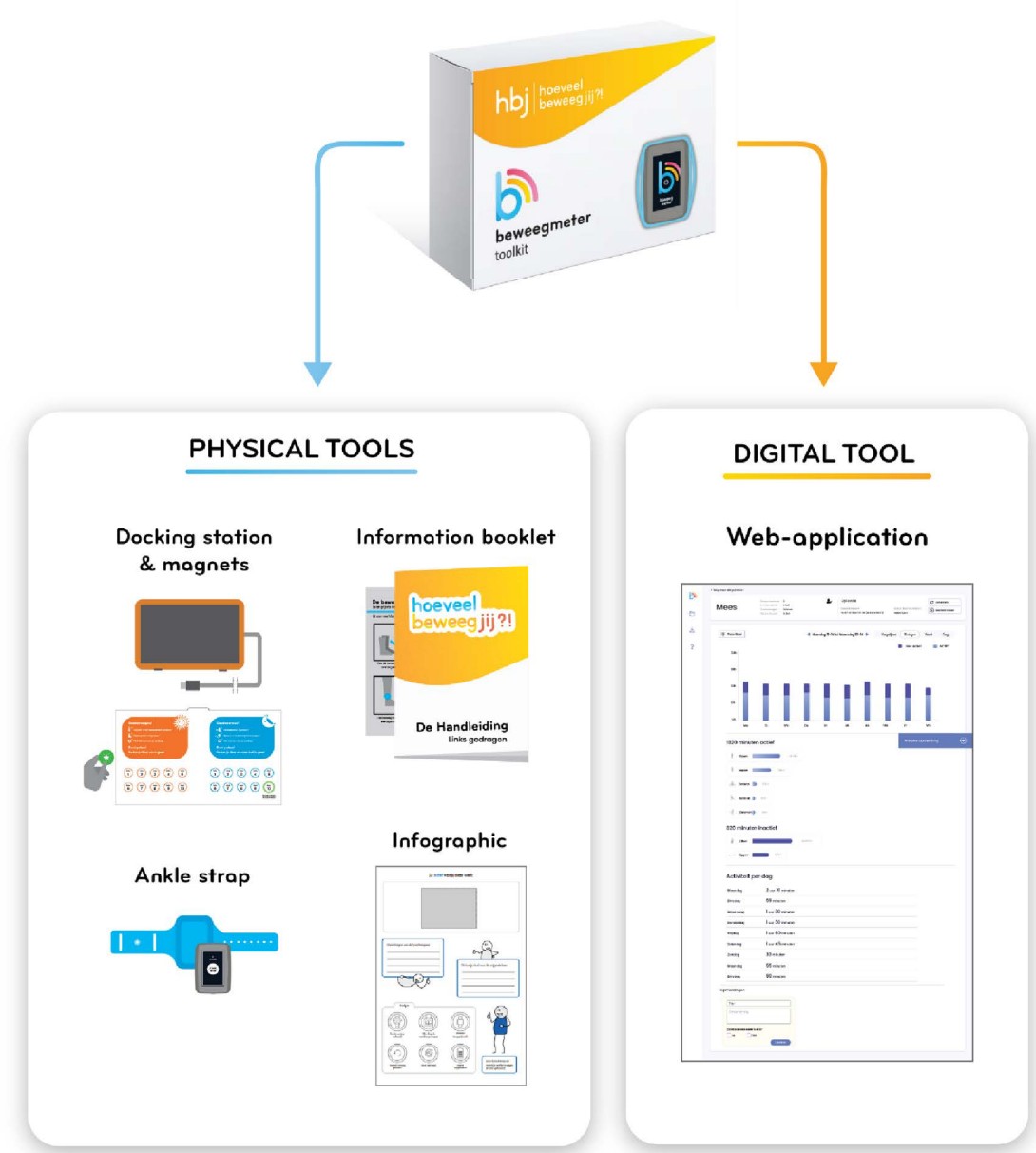

**Fig 3. AM-p Toolkit.** In the center is the AM-p Toolkit depicted, with the four physical tools on the left and the digital tool on the right.

then organized the codes into themes. We used a deductive approach to organize all themes into a tree diagram based on the three areas of feasibility, i.e., acceptability, demand, and practicality. New themes were added inductively if needed. Training data were categorized by 'knowledge,' 'skills,' 'confidence,' and 'content and design of training.' Two independent researchers (BE, AJ, MW, or KR) performed all steps using Atlas.ti (version 9; ATLAS.ti Scientific Software Development GmbH, Berlin, Germany), with a third researcher, consulted if consensus was not reached (MS, MB, or EK). We used manual strategies like mind to merge the results.

## Results

Nineteen PPTs completed the training, but four dropped out during follow-up, leaving 15 to finish the study (Fig 4). Most participants were women (n = 16) aged 36–45, with 84% working in primary care and over 12 years of experience with children (Table 1). PPTs rated their current satisfaction with assessing physical behavior on a 10-point rating scale with a median of 6 [IQR 2]. Most used history taking, diaries, questionnaires, or smartwatches to gather information. Twelve PPTs felt they lacked adequate tools for assessing physical behavior, while five were neutral, and one agreed to have sufficient tools. Dissatisfaction with current assessment methods on a 10-point rating scale (mean score 4) stemmed from issues with patient compliance, subjectivity, and unreliable questionnaires.

Fig 5 outlines the inclusion process for 18 parents and 21 children. Most parents were women aged 36–45 with high education levels. The children had a mean age of 11 years (SD 3.1) and various physical and developmental conditions. Table 2 provides further demographic details of the participants.

### Training

The quantitative data are based on 18 questionnaires and results are shown in Table 3. Qualitative data are based on 2 focus groups, with a total of 14 PPTs participating.

**Knowledge.** On a 5-point Likert scale, most PPTs reported having sufficient knowledge to use the information booklet and docking station. Half of them felt knowledgeable using the web application and infographic.

PPTs appreciated the knowledge conveyed during the training. They mainly said that the knowledge clips on Canvas were self-explanatory and helpful for review. Many indicated that they understood how things worked after the first in-person meeting, with many reporting a need to practice with the web application and infographic to communicate assessment results effectively. Some PPTs valued the training, while others believed good knowledge clips and (online) instructions would suffice, and mandatory training for the AM-p Toolkit might discourage its use.

**Skills.** Almost all PPTs felt skilled in using 'the wearable' and information booklet, and half in using the infographic. Fourteen PPTs felt skilled in using the docking station, and less than half using the web application.

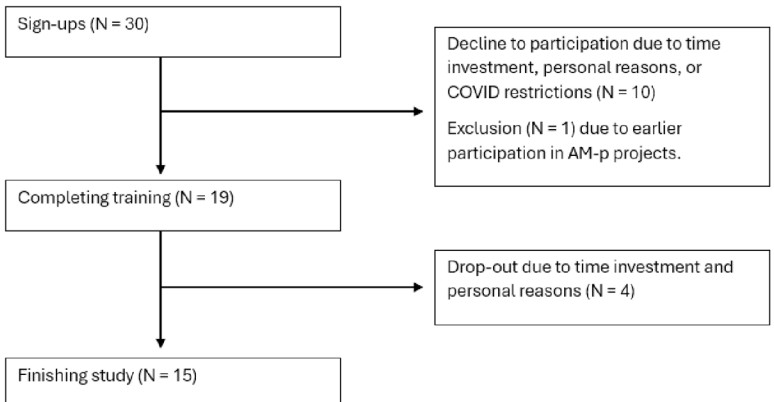

**Fig 4. Flowchart of PPT inclusion procedure.**

**Table 1. Characteristics of included pediatric physiotherapists (PPTs).**

| PPT | Gender | Age group (in years) | Work experience (in years) | Health-care setting (care level) |
|---|---|---|---|---|
| 1 | Wa | 26–35 | 3–5 | Primary |
| 2 | W | 21–25 | 0–2 | Primary |
| 3 | W | 26–35 | 0–2 | Primary |
| 4 | W | 46–55 | >12 | Primary |
| 5 | Mb | 56–65 | >12 | Primary |
| 6 | W | 46–55 | >12 | Primary |
| 7 | W | 36–45 | >12 | Primary |
| 8 | W | 36–45 | >12 | Primary |
| 9 | W | 56–65 | >12 | Primary |
| 10 | W | 26–35 | 3–5 | Secondary |
| 11 | M | 36–45 | 3–5 | Secondary |
| 12 | W | 46–55 | >12 | Secondary |
| 13 | W | 56–65 | >12 | Primary |
| 14 | W | 56–65 | >12 | Primary |
| 15 | W | 46–55 | 3–5 | Primary |
| 16 | W | 36–45 | 6–8 | Primary |
| 17 | M | 26–35 | 9–11 | Primary |
| 18 | W | 21–25 | 0–2 | Primary |
| 19 | W | 36–45 | >12 | Primary |

[a]W woman.

[b]M man.

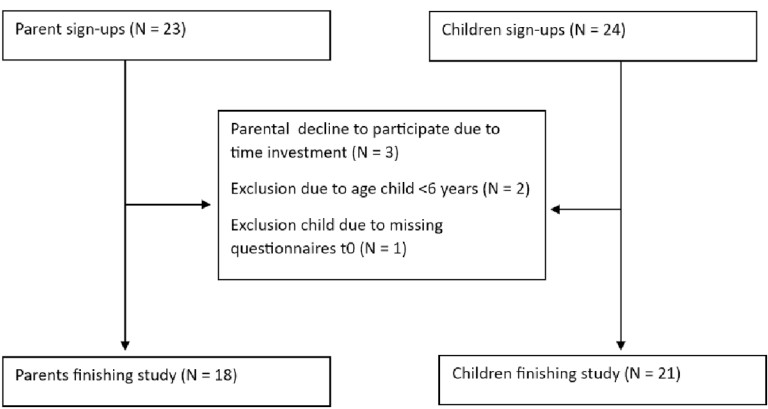

**Fig 5. Flowchart parent and children inclusion procedure.**

Many PPTs felt confident using the AM-p Toolkit and valued the opportunity to practice with professional supervision. However, some needed additional home practice after the first meeting. The main challenge was explaining child-specific assessment results clearly and answering all questions from parents and children. *Focus group 1, PPT#2: "I find no obvious difference between the measurements. I don't find it easy to explain what you see in such a graph and to communicate it objectively to parents. […] the results weren't like with a Movement ABC where you can say, you're in the green zone, you're in the orange zone, you're in the red zone. And this is how you score compared to the norm. So yeah, you must talk your way through it, so to speak."*

**Table 2. Characteristics of included parents and children.**

| Parent | Gender | Age group (in years) | Education level[e] | Child | Gender | Age | Experience with PPT | Diagnosis |
|---|---|---|---|---|---|---|---|---|
| 1 | M[b] | 46–55 | High | 1 | B[c] | 8 | Sometimes | Lyme disease |
| 2 | M | 46–55 | High | 2 | B | 11 | Regularly | Cerebral palsy |
| 3 | W[a] | 36–45 | Middle | 3 | B | 10 | Sometimes | Morbus sever |
| 4 | W | 36–45 | Middle | 4 | B | 14 | Sometimes | Obesity |
| 5 | M | 36–45 | High | 5 | G[d] | 6 | Regularly | Heart problems |
| 6 | W | 36–45 | Low | 6 | G | 6 | Regularly | Dryfus syndrome |
| 7 | W | 46–55 | High | 7 | B | 9 | Regularly | Autism Spectrum Disorder |
| N/A[f] | N/A | N/A | N/A | 8 | G | 17 | Regularly | Lung problems |
| 8 | W | 36–45 | Middle | 9 | B | 11 | Regularly | Obesity |
| 9 | W | 46–55 | High | 10 | B | 14 | Sometimes | Obesity |
| 10 | W | 46–55 | Middle | 11 | B | 11 | Regularly | Obesity |
| 11 | W | 36–45 | High | 12 | G | 10 | First time | No diagnosis |
| 12 | W | 36–45 | Low | 13 | B | 9 | Regularly | Asthma |
| 13 | W | 36–45 | Middle | 14 | B | 10 | Regularly | Obesity |
| 14 | W | 46–55 | High | 15 | B | 15 | Regularly | Pes cavus |
| 15 | W | 36–45 | High | 16 | G | 11 | First time | No diagnosis |
| 16 | W | 26–25 | Middle | 17 | B | 8 | Sometimes | Obesity |
| N/A | N/A | N/A | N/A | 18 | G | 16 | Sometimes | Fatigue |
| 17 | M | 36–45 | Middle | 19 | G | 13 | Regularly | Hip operation |
| N/A | N/A | N/A | N/A | 20 | B | 16 | Regularly | Autism Spectrum Disorder and asthma |
| 18 | W | 36–45 | High | 21 | B | 12 | First time | Obesity |

[a]W woman.

[b]M man.

[c]B boy.

[d]G girl.

[e]Educational level in the Netherlands. Definition of Low: first three years of senior general secondary education and pre-university secondary education; the various pathways of prevocational secondary education, including lower secondary vocational training and assistant's training [38]. Definition of Middle: upper secondary education, basic vocational training, vocational training, and middle management and specialist education [38]. Definition of High: associate degree programs, higher education, Bachelor programs; 4-year education at universities of applied sciences; Master degree programs at universities of applied sciences and research universities; and doctoral degree programs at research universities [38].

[f]N/A: not applicable; participants aged 16 or older did not require parental consent.

**Confidence.** Most PPTs reported confidence in using 'the wearable' and the information booklet in daily work. Thirteen PPTs were confident in using the docking station, and half of them in using the web application and infographic. Only a few PPTs did not feel confident using the AM-p Toolkit in daily practice.

Various PPTs expressed that the in-person meeting was important for developing confidence in using the AM-p Toolkit. They also appreciated hearing about the challenges others faced. Some PPTs felt they needed to be more competent in deciding which group of children the AM-p Toolkit could best use. Overall, PPTs anticipated confidence would grow automatically after having worked a while with the Toolkit: *Focus group 2, PPT#3: "Well, I do think the training is necessary, simply because it gives me more confidence with the Toolkit and everything. So, for me, it helped to work with it; it made me think, okay, this is going to be fine."*

**Content and design of training.** Most PPTs were satisfied with the digital learning environment (CANVAS) and the in-person training. PPTs rated the training (T1 Training) on a 10-point scale with a median of 8 [IQR 2; range 6–10].

**Table 3. T1 (N = 18) questionnaire for the training on a 5-point Likert scale PPTs.**

| Question | Median | IQR | Likert scale options. | | | | |
|---|---|---|---|---|---|---|---|
| | | | 1 | 2 | 3 | 4 | 5 |
| | | | Count of participants who selected each option. | | | | |
| The training met my expectations. | 4 | 1 | – | – | 5 | 10 | 3 |
| After the training, I have sufficient **knowledge** to apply: | | | | | | | |
| 'The wearable' in my field of work. | – | – | – | – | – | – | – |
| The information booklet in my work. | 4 | 1 | – | – | 1 | 9 | 8 |
| The docking station in my work. | 4 | 1 | – | – | 3 | 10 | 5 |
| The web application in my work. | 4 | 1 | – | 1 | 6 | 8 | 3 |
| The infographic in my work. | 4 | 1,25 | – | 2 | 6 | 7 | 4 |
| After the training, I feel sufficiently **skilled** to apply: | | | | | | | |
| 'The wearable' in my work. | 4 | 1 | – | 2 | – | 11 | 5 |
| The information booklet in my work. | 4 | 1 | – | – | 2 | 10 | 6 |
| The docking station in my work. | 4 | 0,5 | – | 1 | 3 | 10 | 4 |
| The web application in my work. | 3 | 1 | – | 2 | 8 | 7 | 1 |
| The infographic in my work. | 4 | 1 | – | – | 8 | 9 | 1 |
| After the training, I have sufficient **confidence** to apply: | | | | | | | |
| 'The wearable' in my work. | 4 | 1 | 1 | 1 | 1 | 10 | 5 |
| The information booklet in my work. | 4 | 1 | – | 1 | 1 | 11 | 5 |
| The docking station in my work. | 4 | 2 | – | 2 | 3 | 8 | 5 |
| The web application in my work. | 4 | 1 | – | 2 | 6 | 8 | 2 |
| The infographic in my work. | 4 | 1 | – | 2 | 6 | 8 | 2 |

Values: 1 – strongly disagree; 2 – disagree; 3 – neutral; 4 – agree; 5 – strongly agree.

IQR: inter quartile range.

PPTs found the knowledge clips supportive and clarifying in the learning environment (CANVAS). PPTs mentioned that parents' understanding of the Toolkit improved after sharing the knowledge clips. Overall, PPTs missed information on analyzing data and applying it to interventions. Most PPTs welcomed the in-person meeting and PPTs agreed that the training duration should be shorter (0,5–1 day).

## AM-p Toolkit

The quantitative data is based on questionnaires (PPTs: n = 15; parents: n = 17; children: n = 20). The qualitative data is based on focus groups among PPTs (n = 14) and individual interviews with parents (n = 11) and children (n = 11). The results are described per domain and theme (Table 4) for PPTs, parents, and children. When they indicate the same, we describe them collectively as end-users.

For PPTs, the overall satisfaction of the AM-p Toolkit on a 10-point scale was 6,3 (SD 1,2). Parents rated the overall satisfaction of the AM-p Toolkit on a 10-point scale with 7,3 (SD 1,6).

**Acceptability.** *Quantitative data.* Six of the PPTs experienced the AM-p Toolkit as helpful in creating a treatment plan, while three did not. Most parents agreed that using the AM-p Toolkit benefited them and their children. Most parents liked that their child wore 'the wearable' and reported that their child successfully wore 'the wearable' for ten days. Most children (n = 13) enjoyed wearing 'the wearable.' When asked if they thought other children liked seeing them with 'the wearable,' the reaction was mostly neutral or positive. See Table 5 for detailed information.

*Qualitative data.* Overall, end-users had mixed experiences with the AM-p Toolkit, with both positive and negative feedback regarding the acceptability of its components.

*Expected added value:* PPTs expected that visualizing and objectifying children's physical behavior embedded in current practices could be beneficial. PPTs expected the AM-p Toolkit to raise awareness, promote shared-decision making, and encourage behavioral changes among parents and children. They also hoped to reduce their workload, enhance the clinical reasoning process, and adapt personalized treatment plans to obtain assessment results. PPTs not accustomed to using advanced technology in their daily routines expected challenges using the AM-p Toolkit. Children and parents were mainly interested in the results of an assessment and their impact on their daily lives.

*Quality and usability:* All end-users found the information booklet to be clear, informative, and easy to follow, with some parents appreciating the additional explanations provided by PPTs for questions. The docking station was generally considered functional, though some found its quality somewhat fragile. Some end-users had difficulties installing it, particularly when connecting it to the internet. The infographic was well-received, offering valuable insights into children's physical behavior patterns during the assessment. Some parents needed additional information to understand the visualizations fully, and PPTs found it challenging to use it to explain results to young children. Children enjoyed the visualized results, finding it fun and engaging. *Child#1: "It was quite funny to see. From about five o'clock to five thirty, we play video games. And on the chart, I saw that I was only active for 1% during this time; that was funny."*

**Table 4. Code tree of qualitative data based on Bowen's domains.**

| Domain | Themes |
|---|---|
| Acceptability | • Expected added value<br>• Quality and usability<br>• Design |
| Demand | • (Non)use<br>• Development AM-p Toolkit<br>• Willingness to future use |
| Practicality | • Time constraints<br>• Integration |

**Table 5. T1 AM-p Toolkit questionnaire with a 5-point Likert scale. Acceptability.**

| | Question | Median | IQR | Likert scale options. | | | | |
|---|---|---|---|---|---|---|---|---|
| | | | | 1 | 2 | 3 | 4 | 5 |
| | | | | Count of participants who selected each option. | | | | |
| PPTs (n = 15) | Using the AM-p Toolkit gave me advantages for creating an individual treatment plan. | 3 | 1 | – | 3 | 6 | 4 | 2 |
| Parents (n = 17) | The use of the AM-p Toolkit provided: | | | | | | | |
| | Me with a personal benefit. | 3 | 1,5 | 1 | 3 | 5 | 6 | 2 |
| | My child with a personal benefit. | 3 | 2 | 1 | 3 | 6 | 3 | 4 |
| | I enjoyed that my child wore 'the wearable.' | 4 | 1 | – | – | 2 | 8 | 7 |
| | My child was able to wear 'the wearable.' for 10 days. | 5 | 0 | 1 | 1 | 1 | 14 | 4 |
| Children (n = 20) | I enjoyed wearing 'the wearable.' | 4 | 1,75 | – | 1 | 6 | 8 | 5 |
| | Other children liked that I wore 'the wearable.' | 3 | 1 | 1 | 1 | 10 | 5 | 3 |

Values: 1 – strongly disagree; 2 – disagree; 3 – neutral; 4 – agree; 5 – strongly agree.

IQR: inter quartile range.

Various end-users reported poor quality of the ankle strap, with issues like it falling off, breaking, or getting damaged. Some children experienced skin irritation and found the strap quickly became dirty and unhygienic. However, not all children and parents encountered problems with the strap. *Parent#12: "It [the strap] broke twice while fastening it. Otherwise, it is a perfect thing. I am impressed that you experience little inconvenience from something like that on your body."*

**Design:** Overall, end-users appreciated the AM-p Toolkit's modern design and child-friendly appearance. The information booklet and docking station received positive reviews, especially. The magnets inside the docking station helped track wearing time, and younger children especially enjoyed them. Most concerns arose about the size and appearance of the ankle strap. While some children and parents found the strap acceptable, others felt insecure or uncomfortable wearing it. *Child#4: "[It was] annoying. [...] they (peers) were nice about it [AM], but I didn't like that they kept asking me what I was wearing."*

PPTs emphasized the importance of considering which children would be comfortable using the AM-p Toolkit. PPTs observed that children and teenagers with obesity were generally less willing to wear the strap, while children with physical disabilities were more accepting of it. *PPT#5: "I saw a 17-year-old girl, she tried it on briefly and said: 'I am definitely not going to wear this.' To her, it looked like an electronic monitoring device [...]. Then I saw a 15-year-old boy who uses orthoses in daily living. He innovatively wore it between the orthotic under his sock, which worked fine."*

**Demand. Quantitative data.** Most PPTs reported that they did not consistently use 'the wearable' with children who met the inclusion criteria. Parents had only positive ratings on understanding how to use the AM-p Toolkit and a positive impression of the insights it provided into their child's physical behavior. All children understood how to use 'the wearable.' Most children also understood how to use the information booklet and the docking station. See Table 6 for detailed information.

**Qualitative data. (Non)use:** PPTs aimed to provide insight for children and parents into individual physical behavior patterns. They were also interested in using the AM-p Toolkit to identify discrepancies between activity patterns and physical capacity or to link physical problems with behavior patterns. PPTs intended to use the AM-p Toolkit with a variety of children,

**Table 6. T1 AM-p Toolkit questionnaire with a 5-point Likert scale. Demand.**

| | Question | Median | IQR | Likert scale options. | | | | |
|---|---|---|---|---|---|---|---|---|
| | | | | 1 | 2 | 3 | 4 | 5 |
| | | | | Count of participants who selected each option. | | | | |
| PPTs (n = 15) | I have always used 'the wearable' for children who met the inclusion criteria. | 2 | 1 | – | 8 | 5 | 2 | – |
| Parents (n = 17) | I understood how to use the: | | | | | | | |
| | 'the wearable.' | 5 | 0,5 | – | – | 3 | 1 | 13 |
| | Information booklet. | 5 | 1,5 | – | – | 4 | 2 | 11 |
| | Docking station. | 5 | 1 | – | – | 3 | 2 | 12 |
| | The AM-p Toolkit provided insight into my child's physical behavior. | 4 | 1 | – | – | 2 | 8 | 7 |
| Children (n = 20) | I understood how to use the: | | | | | | | |
| | 'the wearable.' | 5 | 1 | – | – | 1 | 7 | 12 |
| | Information booklet. | 5 | 1 | 1 | 3 | 3 | 13 | 1 |
| | Docking station. | 4.5 | 1,75 | 1 | – | 4 | 5 | 10 |

Values: 1 – strongly disagree; 2 – disagree; 3 – neutral; 4 – agree; 5 – strongly agree.

IQR: inter quartile range.

including those with medically unexplained symptoms, obesity, or multiple disabilities. However, some children, such as those with sensory issues, autism, or those uncomfortable with the visibility of 'the wearable,' refused to wear it. PPTs felt that the data provided by the AM-p Toolkit did not always align with their intervention strategies (e.g., assessing the intensity of physical activity).

*Development Toolkit:* A few PPTs raised concerns about the accuracy of the output, being concerned that 'the wearable' could not distinguish between actual activity and simple movements like jiggling a foot. End-users desired more detailed insights into activity intensity, such as heart rate monitoring. *Child#5: "Yeah, as far as I'm concerned, there is a big difference between moving with a heartbeat of 130 or going back and forth in the kitchen with a heartbeat of 80."*

Many end-users were interested in further developing the AM-p Toolkit into an intervention tool with live-tracking options, allowing parents and children to see daily activity results immediately. PPTs and parents also emphasized the need for a helpdesk to address technical issues. PPTs preferred a mobile version of the web application for quicker access than logging onto a computer. Additionally, PPTs expressed the need for reference values to interpret the results better and integrate them into treatment plans. They struggled to analyze children's physical behavior patterns and found it challenging to translate the assessment data into meaningful goals. While some parents were content with general recommendations, such as "being more active after 14:30," others required more specific guidance on incorporating activity into daily routines. *Parent#12: She [PPT] advised trying to move for more than 300 minutes a day. But well, I thought, that is quite a lot per day, right?"*

To make the AM-p Toolkit more inclusive, PPTs and parents requested support in languages other than Dutch.

*Willingness to future use:* PPTs identified the ideal target groups for using the AM-p Toolkit as children with developmental disabilities or those whose activity levels do not match their physical capabilities. While some PPTs wanted to use the AM-p Toolkit immediately to evaluate basic activities, others preferred to wait for further development. *PPT#8: "I definitely think this [AM-p Toolkit] adds value for measuring physical activity. Of course, everything can be improved and developed, but I believe this is a good foundation. Yes, I would apply it in practice."*

Many parents recognized their child's physical behavior pattern in the results and were enthusiastic about their new insights, while children's opinions varied. *Child#16: "Yes, because I found it to be a nice experience to wear ['the wearable'], and after a while, I became less self-conscious about it, so I would like to wear it again."*

**Practicality. *Quantitative data.*** Overall, PPTs rated the AM-p Toolkit as making their work easier, with mainly neutral to positive feedback. However, a few negative ratings were given for the docking station and web application. Integrating the AM-p Toolkit in practice received varied feedback: the information booklet and infographic were rated positively, while a small number of negative ratings were given for 'the wearable,' docking station, and web application. Nearly all PPTs felt confident introducing the wearable and providing feedback on the results, with only one reporting a lack of confidence.

Overall, the information provided by the PPT and the information booklet was sufficient for wearing 'the wearable.' Most parents found 'the wearable' practical, though a few were not convinced of its practicality. Regarding compatibility with everyday activities, parents and children reported that 'the wearable' worked well except for sports activities. See Table 7 for detailed information.

*Qualitative data.    Time constraints*: Using the AM-p Toolkit required more time than standard care, as PPTs provided practical help and guidance to parents who experienced problems

**Table 7. T1 AM-p Toolkit questionnaire with a 5-point Likert scale. Practicality.**

| | Question | Median | IQR | Likert scale options. | | | | |
|---|---|---|---|---|---|---|---|---|
| | | | | 1 | 2 | 3 | 4 | 5 |
| | | | | Count of participants who selected each option. | | | | |
| PPT (n = 15) | My work was easier by using the: | | | | | | | |
| | 'the wearable.' | 3 | 1 | – | 3 | 8 | 4 | – |
| | Information booklet. | 4 | 2 | – | – | 5 | 5 | 5 |
| | Docking station. | 3 | 1 | 1 | 1 | 8 | 4 | 1 |
| | Web application. | 4 | 2 | – | 1 | 5 | 5 | 4 |
| | Infographic. | 3 | 2 | – | – | 8 | 3 | 4 |
| | The way I am used to work integrates well with the: | | | | | | | |
| | 'the wearable.' | 3 | 1 | 2 | 1 | 6 | 4 | 2 |
| | Information booklet. | 4 | 1 | – | – | 6 | 6 | 3 |
| | Docking station. | 3 | 1 | 1 | 3 | 8 | 2 | 1 |
| | Web application. | 4 | 2 | 1 | – | 5 | 5 | 4 |
| | Infographic. | 4 | 2 | – | – | 6 | 5 | 4 |
| | I felt sufficiently skilled to provide | | | | | | | |
| | Children and parents with an introduction to the AM-p Toolkit. | 4 | 1 | 1 | – | 3 | 8 | 3 |
| | Children and parents with feedback on an assessment. | 3 | 1 | 1 | – | 7 | 7 | – |
| Parents (n = 17) | The information provided by: | | | | | | | |
| | The PPT was sufficient to wear 'the wearable.' | 5 | 0,5 | – | 1 | 2 | 1 | 13 |
| | The information booklet was sufficient to wear 'the wearable.' | 5 | 1,5 | – | – | 4 | 1 | 12 |
| | It was practical to use the: | | | | | | | |
| | 'the wearable.' | 4 | 2,5 | – | 4 | 3 | 4 | 6 |
| | Information booklet. | 4 | 1 | 1 | – | 2 | 7 | 7 |
| | Docking station. | 4 | 1,5 | – | – | 4 | 8 | 5 |
| | My child was able to do all activities with 'the wearable:' | | | | | | | |
| | At home. | 5 | 1 | – | – | – | 5 | 12 |
| | At school. | 5 | 1 | – | – | 1 | 5 | 11 |
| | During play. | 5 | 1 | – | – | – | 6 | 11 |
| | During sports. | 4 | 1,5 | – | 3 | 1 | 5 | 8 |
| Children (n = 20) | I was able to wear 'the wearable' as my PPT explained to me. | 5 | 1 | – | – | – | 7 | 13 |
| | I was able to do all activities with 'the wearable' on my ankle: | | | | | | | |
| | At home. | 5 | 1 | – | – | – | 6 | 14 |
| | At school. | 5 | 1 | – | – | – | 6 | 14 |
| | During play. | 5 | 1 | – | – | 1 | 6 | 13 |
| | During sports. | 5 | 1 | 2 | 1 | 1 | 5 | 11 |

Values: 1 – strongly disagree; 2 – disagree; 3 – neutral; 4 – agree; 5 – strongly agree.

IQR: inter quartile range.

using the concomitant tools at home. PPTs offered practical help and advice to parents in a way that required a significant amount of time. They reported frequently checking data uploads to ensure accuracy and update parents. Some PPTs made choices based on the time investment needed rather than the necessity of assessing a child. *PPT#3: "I would give the AM less likely to a teenager who is disorganized and whose caretaker is not engaged in therapy because then, I think, I'd have to monitor [the teenager] to work it out closely. […] it is quite determining in 'how important do I find the result that I am going to achieve [with this assessment]'?"*

*Integration:* The quality of the ankle strap of 'the wearable' posed challenges for integrating the AM-p Toolkit into current workflows, particularly for children and parents. Depending on sports, some activities could not be assessed, as 'the wearable' needed to be waterproof and fit securely over gear. In contrast, the information booklet and knowledge clips easily fit into existing processes. Connecting the docking station to the internet was sometimes difficult, leaving some parents uncertain about proper installation. They often sought help from others or their PPT for installation. However, once the initial connection was made and data upload was confirmed, parents reported no issues connecting the strap to the docking station. Using the infographic to explain results to parents and children was occasionally challenging for PPTs. PPT#4: "*I said that he wasn't moving enough. But I did not see it [in the output]; I even thought he moved quite a bit. However, I told him we needed to get those activity lines a bit higher, but I found that very difficult. I didn't know how to sell it, so to speak.*"

## Discussion

The objectives of our study are 1) To explore the feasibility (acceptability, demand, practicality) of the AM-p Toolkit in PPT practice from the perspectives of PPTs, children, and parents to gather feedback for further refinement, and 2) To evaluate whether the content and design of the training align with the needs of PPTs and assess how the training affects their knowledge, skills, and confidence using the AM-p Toolkit. PPTs, parents, and children perceived the AM-p Toolkit positively, particularly its value in facilitating awareness of and visualizing physical behavior. PPTs were optimistic about future use and gave valuable input for further development. Data revealed that the training in its current form largely meets the needs of PPTs. While most PPTs gained sufficient knowledge, skills, and confidence in using the tools, differences between the tools were present.

Before joining the study, PPTs expressed dissatisfaction with their current methods for assessing physical behavior and saw potential value in the AM-p Toolkit. However, for many PPTs and parents, the actual benefits based on assessment results remained unclear. This is a well-known barrier because clinicians work in a hectic clinical environment and do not recognize the value of the critical information that wearables can obtain [25]. Furthermore, they struggle to implement technology in their working routines [25]. The design and quality of the hardware still need further development. The design of the strap was based on input from parents and children in a previous study, including parents of children who use a wheelchair in daily life, who reported that 'the wearable' should be compatible with all activities in daily life and should be able to withstand collisions [27]. Further development should focus on creating a smaller, less obtrusive design. There is certainly demand from all end-users, provided the AM-p Toolkit is improved. Most practicality issues stemmed from (1) using prototypes, which required additional time, and (2) a missing translation of the implication of assessment results for clinical practice and individual, meaningful goal-setting. In line with the development phase of the Technology Readiness Level Scale, our study includes the end-users and health perspectives. Next, the focus should be on quick, design processes in which the technology and concurrent tools are optimized. It is essential to maintain a close relationship between end-users and technical factors during the design process and to incorporate different field experts in co-design and co-evaluation in iterative processes [30,39].

PPTs mentioned that they were interested in using the AM-p Toolkit to indicate a potential discrepancy of load and capacity or connecting physical problems to activity patterns. Insight into the intensity of activities will add to the demand for activity monitoring in PPT practice. However, the search for the best method of gauging both the type and intensity of physical activity in children is still ongoing, and models are far from ready to be used in free-living settings, let alone for use in the pediatric population with motor impairments [40]. Therefore,

the output of the AM-p has to be developed (e.g., intensity or step counts), and the training should be aligned with the output developments and translations from theory to practice. One may wonder if another new activity monitoring instrument is necessary in the crowded market. To our knowledge, there is no single, practical device that can assess the physical behavior of children, including those with developmental disabilities and those who use a wheelchair. Furthermore, recent research found that the role of healthcare professionals is to convey the complexities of device-based assessments in terms that are easily understandable for the patient when interpreting assessment results and evaluate personalized activity goals [41]. Proper training for health care professionals is essential for confidentially using this skill. Moreover, a PPT should focus on individually tailored approaches to enhance the activity and participation within the possibilities of a child in their environment. The AM-p Toolkit can help support the clinical reasoning process and evaluate the physical behavior over time.

Overall, PPTs were satisfied with the training. After completing the training, a few PPTs felt knowledgeable, skilled, and confident in using the web application and infographic. Various PPTs reported a need to practice with the web application and infographic to communicate assessment results effectively. That is hardly surprising, as interpreting activity patterns, translating results into practical advice, and integrating assessment results into a treatment plan is complex as there are no reference values. Children and parents, however, indicated that the graphs provided insight into the movement patterns.

The fact that PPTs need extra practice using the web application is in line with other research indicating that therapists find it challenging to incorporate technology in daily practice due to a lack of knowledge and ability to apply the technology themselves and to their patients or because they find it too expensive or time-consuming [42]. The web application was informative; however, PPTs needed help integrating it into current practices. Logging in was time-consuming, and PPTs preferred using a mobile app instead of a laptop or desktop computer.

The training had the smallest impact on improving confidence. We saw a wide variation in confidence regarding the use of different tools. Some therapists reported that the training needed to give them more confidence to use specific tools in PPT practice. The impact of training on knowledge and skills was more apparent. For instance, each therapist reported having sufficient knowledge and skills in using the ankle strap, information booklet, and docking station. Therapists can feel skilled but still need more confidence in using the tools. Confidence often grows with practice and familiarity. Recent studies shed light on perfectionism, imposter phenomenon, and mental health in medical professions [43,44]. The imposter phenomenon is highly prevalent among licensed physical therapists [44], which means that individuals doubt their abilities and fear being accused of fraud. We suggest that even if individuals are objectively capable and skilled, their perception and attitude can undermine their confidence. Literature shows that PTs perceive activity monitoring as useful, with the current use of wearable technology determined by the individual therapist's skills, beliefs, and attitudes [45].

Although PPTs were satisfied with the blended training approach, they were less satisfied with the study workload. Most PPTs found the training too extensive and proposed a more efficient setup. Although previous studies indicate that extensive training is necessary to feel competent in using newly developed tools [27,46,47], we recommend to examine the study workloadcritically. We believe that adequate training is essential for both efficiency and effectiveness, but it is equally important that sufficient time is allocated to achieve meaningful training outcomes. A facilitating role from employers, alongside the accreditation of training, could significantly contribute to this process.

PPTs reported that they could not include all children they thought suitable for the study because they refused to wear 'the wearable.' It remains uncertain whether participating PPTs

genuinely approached all children for participation or if certain assumptions were made in advance regarding their interest or suitability. PPTs may have preemptively determined that a particular child or their parents would not be interested or that participation would not be appropriate. Such assumptions, whether intentional or not, can have significant implications for inclusivity and representation in studies, potentially limiting diverse participation and introducing subtle biases into the selection process. We have obtained new insights that can be used to develop the AM-p Toolkit further through iterative steps [48]. Additionally, we should reconsider how the AM-p Toolkit fits into the broader workflow of therapy sessions, ensuring that time remains a critical aspect of the development process. We acknowledge that our statistical analyses could have been more conclusive with a larger number of participants. Therefore, we advise to design future studies with more participants to allow for better statistical analyses.

We opted to use self-developed questionnaires, which were not validated, as our primary goal was to gather specific information about the individual tools rather than the AM-p Toolkit. Using a validated questionnaire, such as the System Usability Scale [21,49] or the Technology Acceptance Model [50,51], would have provided a valid outcome to assess the overall feasibility of the AM-p Toolkit. However, these questionnaires have challenges: they are often too complex for children to understand and would require significant adaptation. Moreover, the Technology Acceptance Model consists of eight subscales, which would have been overly burdensome for participants.

## Future directions

Iterations are necessary to develop the AM-p Toolkit further, understand why its use is time-intensive, and how this affects the actual use in practice [48]. A potential quick adaptation could involve adding QR codes for the knowledge clips to the docking station. Originally designed for training PPTs, these clips explained how the AM-p Toolkit functions. PPTs and parents found these clips helpful when using the AM-p Toolkit, enhancing their understanding and effective use. Quick action is crucial because eHealth interventions and their evaluations rapidly become outdated due to rapid technological and medical advancements [52]. However, the rise of nontraditional developers producing a wide range of online resources makes it harder to identify scientifically validated interventions. We must acknowledge and adapt to this fast-paced, ever-changing technological, science, and eHealth landscape [52].

Future development should aim to test modifications that reduce time while maintaining efficiency, i.e., by exploring whether parts of the AM-p Toolkit can be automated or supported by technology to reduce the time burden on PPTs. We also recommend a helpdesk for technical support and parental assistance so that PPTs do not have to spend time troubleshooting.

## Conclusion

The AM-p Toolkit holds promise in facilitating awareness and visualizing physical behavior. However, further development is needed to make all tools fully suitable for daily PPT practice. The acceptability was generally positive among PPTs and children, whereas mixed for parents who missed the personal benefit of an assessment. Regarding the demand, PPTs faced challenges using the toolkit for all eligible children, citing issues with compliance and missing physical activity intensity parameters. Parents and children understood the toolkit's use and valued the insights into the child's physical behavior. For better practicality, the web application requires further development to enhance ease of use. Refinement of the AM-p and its strap is essential to ensure stable assessment results and a higher wearing comfort. The training is of added value and generally necessary for gaining knowledge, skills, and confidence in

using the AM-p Toolkit. The content should focus more on analyzing assessment results and embedding them in the clinical reasoning process as well as functional goal-setting. Also, the duration of training should be critically evaluated to better align with the perceived needs of PPTs, as it remains questionable whether a shorter training would ensure a smooth and feasible daily use of the AM-p Toolkit.

## Acknowledgments

We thank all children, their parents or caregivers, and all PPTs who participated in the study. Furthermore, we thank Rins Rutgers for data curation of AM-p data and software development of the web application); Lotte van der Schoot for investigation during the co-design process and the visualization of Fig 3); Jelle Rijpkema for investigation during the co-design process and the development of the web application); Marry Bassa for investigation during the co-design process; Anne Zeegers for investigation during the co-design process; students of the Master's Program of Pediatric Physical Therapy of the University of Applied Sciences, Utrecht: Anne Jansen, Marije Wiegersma, Kelly Rotteveel for their support during the formal analysis of data, investigation, provision of resources. We thank Judy van Es and Daniëlle Kortlever for their support during the formal analysis. We also thank Dr. Corelien Kloek for her input on methodology.

## Author contributions

**Conceptualization:** Barbara Engels, Elles Kotte, Raoul Engelbert, Marleen Sol, Remko van der Lugt, Harriët Wittink, Jan Willem Gorter, Manon A. T. Bloemen.

**Data curation:** Barbara Engels, Elles Kotte.

**Formal analysis:** Barbara Engels, Elles Kotte, Marleen Sol.

**Funding acquisition:** Manon A. T. Bloemen.

**Investigation:** Barbara Engels, Elles Kotte, Marleen Sol, Remko van der Lugt, Manon A. T. Bloemen.

**Methodology:** Barbara Engels, Elles Kotte, Raoul Engelbert, Marleen Sol, Remko van der Lugt, Harriët Wittink, Jan Willem Gorter, Manon A. T. Bloemen.

**Project administration:** Barbara Engels, Manon A. T. Bloemen.

**Supervision:** Barbara Engels, Elles Kotte, Marleen Sol, Jan Willem Gorter, Manon A. T. Bloemen.

**Validation:** Barbara Engels, Elles Kotte, Raoul Engelbert, Marleen Sol, Jan Willem Gorter, Manon A. T. Bloemen.

**Visualization:** Barbara Engels, Elles Kotte.

**Writing – original draft:** Barbara Engels, Elles Kotte, Jan Willem Gorter, Manon A. T. Bloemen.

**Writing – review & editing:** Raoul Engelbert, Marleen Sol, Remko van der Lugt, Harriët Wittink.

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
