## [Decision Letter · Decision Letter 0]

13 Jan 2025

PONE-D-24-52313Monitoring physical behavior in pediatric physical therapy: A mixed methods feasibility study to evaluate a newly developed toolkit and trainingPLOS ONE

Dear Dr. Engels,

Thank you for submitting your manuscript to PLOS ONE. After careful consideration, we feel that it has merit but does not fully meet PLOS ONE’s publication criteria as it currently stands. Therefore, we invite you to submit a revised version of the manuscript that addresses the points raised during the review process.

We look forward to receiving your revised manuscript.

Kind regards,

Karthikeyan Thiyagarajan PhD

Academic Editor

PLOS ONE

Additional Editor Comments:

Dear Authors,

On behalf of the Editorial Board, we thank you for submitting your manuscript to PLOS ONE. Reviewers have now completed their reviews of your manuscript, and one of the reviewers suggested a minor revision.

Please carefully read the comments and suggestions from reviewers and respond to each comment and revise the manuscript accordingly. Please submit the revised manuscript with both marked and unmarked versions by 3rd February 2025.

I have some suggestions and comments:

I appreciate your work concerning the development of an activity monitoring prototype toolkit (AM-p Toolkit) using the Bowen framework that helps pediatric physical therapists to

assess the physical activity of children with the data obtained. However, the study used fewer participants; hence, I suggest using more participants in your future study for better statistical analysis.

Please don't use an abbreviation the first time; for instance, an abbreviation, PA, was presented in the abstract. I presume it was for physical activity. Please state directly Physical Activity if it is for PA.

You could represent the statistical data/results using appropriate statistical graphs and figures; for instance, the data distribution can be graphically presented.

Was the distribution of the skewed data positively skewed or negatively skewed? Have you done the analysis after the transformation of data, like log transformation, etc.?

Why didn't you choose healthy children as a control to compare the children with health issues as you have indicated in Table 2?

Please indicate how this kind of study will help to improve the skills, cognitive ability, and quality of life of disease-affected children as that of normal children.

I found some typos; please make the results and discussion contents more precise. 

Yours sincerely,

Karthikeyan Thiyagarajan PhD

Academic Editor

PLOS ONE

Reviewers' comments:

Reviewer's Responses to Questions

**Comments to the Author**

1. Is the manuscript technically sound, and do the data support the conclusions?

Reviewer #1: Yes

Reviewer #2: Yes

2. Has the statistical analysis been performed appropriately and rigorously? 

Reviewer #1: Yes

Reviewer #2: Yes

3. Have the authors made all data underlying the findings in their manuscript fully available?

Reviewer #1: Yes

Reviewer #2: Yes

4. Is the manuscript presented in an intelligible fashion and written in standard English?

Reviewer #1: Yes

Reviewer #2: Yes

5. Review Comments to the Author

Reviewer #1: the authours present a succint description of their study. The topic is relavant and provides deeper appraciation of the topic and gap. Below are some concerns

if children assent was not done. Kindly consider this in future.

some tables have empty spaces and incomplete information such as Table 2 Characteristics of included parents and children.

Reviewer #2: Dear Author,

The work you and your co-authors have done is truly commendable. However, I have a few queries outlined below:

The objectives are still not clearly defined.

Regarding "parents," does this refer to the involvement of both father and mother? Please clarify.

6. PLOS authors have the option to publish the peer review history of their article (what does this mean? ). If published, this will include your full peer review and any attached files.

**Do you want your identity to be public for this peer review?** For information about this choice, including consent withdrawal, please see our Privacy Policy .

Reviewer #1: **Yes: ** Gideon Awenabisa Atanuriba

Reviewer #2: **Yes: ** Dr Sharath Hullumani

---

## [Author Response · Author response to Decision Letter 1]

21 Jan 2025

Dear editor, dear Dr. Karthikeyan Thiyagarajan,

Thank you for the opportunity to revise and resubmit our manuscript, “Monitoring physical behavior in pediatric physical therapy: A mixed methods feasibility study to evaluate a newly developed toolkit and training.” We appreciate your time and constructive feedback provided by you and the reviewers, which has strengthened our work. Below, we have addressed each remark in detail, highlighting the changes made in the manuscript.

Editor

Remark 1: I appreciate your work concerning the development of an activity monitoring prototype toolkit (AM-p Toolkit) using the Bowen framework that helps pediatric physical therapists to assess the physical activity of children with the data obtained. However, the study used fewer participants; hence, I suggest using more participants in your future study for better statistical analysis.

Answer to remark 1: we added this suggestion in the discussion section of the manuscript (p 29, line 619): “We acknowledge that our statistical analyses could have been more conclusive with a larger number of participants. Therefore, we advise to design future studies with more participants to allow for better statistical analyses.”

Remark 2: Please don't use an abbreviation the first time; for instance, an abbreviation, PA, was presented in the abstract. I presume it was for physical activity. Please state directly Physical Activity if it is for PA.

Answer to remark 2: We adjusted the abbreviation in the abstract, it was an unintentional oversight.

Remark 3: You could represent the statistical data/results using appropriate statistical graphs and figures; for instance, the data distribution can be graphically presented.

Answer to remark 3: Thank you for this comment. It was our initial intention to represent the data with graphs. However, as the questionnaires of PPTs, parents and children were not completely identical, this would have resulted in 9 graphs for PPTs, 8 graphs for parents, and 5 graphs for children, a total of 22 graphs. After discussion with the team, we decided to depict the data in tables. This was therefore a practical choice to ensure that data could be presented clearly.

Remark 4: Was the distribution of the skewed data positively skewed or negatively skewed? Have you done the analysis after the transformation of data, like log transformation, etc.?

Answer to remark 4: The skewed data distribution was mostly positively skewed. For this study, we chose to present IQRs along with absolute values because they offer a clear and straightforward summary of the data, making the results more accessible and interpretable. This approach aligns with our goal of clearly communicating findings to a broad range of end-users.

We do, however, see merit in the suggestion to use log transformations in future research, especially when evaluating the results of a larger group of participants. Such transformations could provide additional benefits by stabilizing variability, reducing skewness, and enable more nuanced comparisons across subgroups.

Remark 5: Why didn't you choose healthy children as a control to compare the children with health issues as you have indicated in Table 2?

Answer to remark 5: Thank you for raising this point. The aim of the study was to evaluate the current feasibility of the AM-p Toolkit in a real-life setting in which PPTs could independently select participants they saw merit in assessing with the AM-p Toolkit in the period between February and May 2022. Based on outcomes of this study, there is further development needed of the AM-p Toolkit. At this point, comparing results with healthy controls would not be appropriate. We agree that future studies, after further developing the toolkit, could benefit from comparing children with and without developmental disabilities to evaluate e.g., whether one group derives greater advantage from the AM-p Toolkit.

Remark 6: Please indicate how this kind of study will help to improve the skills, cognitive ability, and quality of life of disease-affected children as that of normal children.

Answer to remark 6: This study is situated in the developmental process of the AM-p Toolkit. With the insights gained, we aim to refine the toolkit further to meet the different needs of its end-users, making it a suitable assessment tool for pediatric physiotherapy practice. Such studies help us improve insight into children's objectively measured physical behavior. This supports the integration of assessment results into the clinical reasoning process, meaningful goal-setting, and individually tailored therapy for children with and without developmental disabilities. Over time, this can help improve the physical activity levels of all children, which ultimately supports a better quality of life and cognitive and physical development.

Remark 7: I found some typos; please make the results and discussion contents more precise.

Answer to remark 7: We have reviewed these sections again for clarity, correcting typos and refining the language where necessary (see track changes manuscript). If needed, please let us know if there are any further adjustments necessary.

Reviewer #1:

Remark 1: if children assent was not done. Kindly consider this in future.

Answer to remark 1: Thank you for this important remark. We asked for children’s assent to participate but described the obtained written consent only. We corrected the description of the procedure accordingly (p 8, line 140): “PPTs, children, parents or caregivers received written and verbal information about the study, along with an informed consent form. Written informed consent was obtained from PPTs and parents or caregivers. For children under 12, assent was obtained from the child, and parents or caregivers provided written consent. For children aged 12 to 16, both assent and separate written consent from parents or caregivers were given. From age 16, written consent was obtained directly from the child, provided they had appropriate cognitive development.”

Remark 2: some tables have empty spaces and incomplete information such as Table 2 Characteristics of included parents and children.

Answer to remark 2: The empty spaces in Table 2 reflect the fact that some of the participating children were 16 years or older and, therefore, did not require parental consent to participate. In these cases, the children independently made the decision to participate in the study. For clarification, we added “N/A” in the table with an explanation in the legend: “fN/A: not applicable; participants aged 16 or older did not require parental consent.”

Reviewer #2:

Remark 1: The objectives are still not clearly defined.

Answer to remark 1: To ensure the objectives are clearly defined, we applied changes to the last paragraph of our introduction. Applied changes (p 5, line 75): “The objectives of our study are 1) To explore the feasibility (acceptability, demand, practicality) of the AM-p Toolkit in PPT practice from the perspectives of PPTs, children, and parents to gather feedback for further refinement, and 2) To evaluate whether the content and design of the training align with the needs of PPTs and assess how the training affects their knowledge, skills, and confidence using the AM-p Toolkit.”

As a logical consequence of these adjustments, we have also adapted the conclusions to align with the objectives and changed the abstract accordingly.

Discussion (p 26, line 522): “The objectives of our study are 1) To explore the feasibility (acceptability, demand, practicality) of the AM-p Toolkit in PPT practice from the perspectives of PPTs, children, and parents to gather feedback for further refinement, and 2) To evaluate whether the content and design of the training align with the needs of PPTs and assess how the training affects their knowledge, skills, and confidence using the AM-p Toolkit. PPTs, parents, and children perceived the AM-p Toolkit positively, particularly its value in facilitating awareness of and visualizing physical behavior. PPTs were optimistic about future use and gave valuable input for further development. Data revealed that the training in its current form largely meets the needs of PPTs. While most PPTs gained sufficient knowledge, skills, and confidence in using the tools, differences between the tools were present.”

Conclusions (p 31, line 631): “The AM-p Toolkit holds promise in facilitating awareness and visualizing physical behavior. However, further development is needed to make all tools fully suitable for daily PPT practice. The acceptability was generally positive among PPTs and children, whereas mixed for parents who missed the personal benefit of an assessment. Regarding the demand, PPTs faced challenges using the toolkit for all eligible children, citing issues with compliance and missing physical activity intensity parameters. Parents and children understood the toolkit’s use and valued the insights into the child’s physical behavior. For better practicality, the web application requires further development to enhance ease of use. Refinement of the AM-p and its strap is essential to ensure stable assessment results and a higher wearing comfort. The training is of added value and generally necessary for gaining knowledge, skills, and confidence in using the AM-p Toolkit. The content should focus more on analyzing assessment results and embedding them in the clinical reasoning process as well as functional goal-setting. Also, the duration of training should be critically evaluated to better align with the perceived needs of PPTs, as it remains questionable whether a shorter training would ensure a smooth and feasible daily use of the AM-p Toolkit.”

Abstract (p 2, line 20): “The AM-p Toolkit shows promise in PPT, with most end-users seeing potential for its use in the future to assess physical behavior in children. PPTs see potential for certain groups of children who can benefit from the AM-p Toolkit. Development should focus on improvements of the digital tool and refinement of the activity monitor’s fixation. Training is important and can be strengthened by emphasizing the analysis of assessment results, clinical reasoning, and functional goal-setting.”

Remark 2: Regarding "parents," does this refer to the involvement of both father and mother? Please clarify.

Answer to remark 2: In our study, the term “parents” refer to the involvement of either the father or the mother, as participating caregivers were given the autonomy to decide which one of them would take part. We acknowledge that including both parents in future studies could provide deeper insights into the potential benefits and challenges of using assessment tools like the AM-p Toolkit in a child’s activities.

---

## [Decision Letter · Decision Letter 1]

6 Feb 2025

Monitoring physical behavior in pediatric physical therapy: A mixed methods feasibility study to evaluate a newly developed toolkit and training

PONE-D-24-52313R1

Dear Dr. Engels,

We’re pleased to inform you that your manuscript has been judged scientifically suitable for publication and will be formally accepted for publication once it meets all outstanding technical requirements.

Kind regards,

Karthikeyan Thiyagarajan PhD

Academic Editor

PLOS ONE

Additional Editor Comments (optional):

Dear Authors,

After careful scientific evaluations with peer reviews, I am pleased to confirm the manuscript entitled "Monitoring physical behavior in pediatric physical therapy: A mixed methods feasibility study to evaluate a newly developed toolkit and training." has been accepted for publication in PLOS ONE.

Kind regards,

Karthikeyan Thiyagarajan PhD

Academic Editor, PLOS ONE.

Reviewers' comments:

Reviewer's Responses to Questions

**Comments to the Author**

1. If the authors have adequately addressed your comments raised in a previous round of review and you feel that this manuscript is now acceptable for publication, you may indicate that here to bypass the “Comments to the Author” section, enter your conflict of interest statement in the “Confidential to Editor” section, and submit your "Accept" recommendation.

Reviewer #1: All comments have been addressed

2. Is the manuscript technically sound, and do the data support the conclusions?

Reviewer #1: Yes

3. Has the statistical analysis been performed appropriately and rigorously? 

Reviewer #1: Yes

4. Have the authors made all data underlying the findings in their manuscript fully available?

Reviewer #1: Yes

5. Is the manuscript presented in an intelligible fashion and written in standard English?

Reviewer #1: Yes

6. Review Comments to the Author

Reviewer #1: (No Response)

7. PLOS authors have the option to publish the peer review history of their article (what does this mean? ). If published, this will include your full peer review and any attached files.

**Do you want your identity to be public for this peer review?** For information about this choice, including consent withdrawal, please see our Privacy Policy .

Reviewer #1: **Yes: ** Gideon Awenabisa Atanuriba

---

## [Editor Report · Acceptance letter]

PONE-D-24-52313R1

PLOS ONE

Dear Dr. Engels,

I'm pleased to inform you that your manuscript has been deemed suitable for publication in PLOS ONE. Congratulations! Your manuscript is now being handed over to our production team.

Kind regards,

on behalf of

Dr. Karthikeyan Thiyagarajan

Academic Editor

PLOS ONE